# Application of Data Envelopment Analysis (DEA) in the Selection of Sustainable Suppliers: A Review and Bibliometric Analysis

**Katerina Fotova Čiković [1],*** , **Ivana Martinčević [2]** and **Joško Lozić [1]**

1   Department of Economy, University North, Trg dr. Žarka Dolinara 1, 48000 Koprivnica, Croatia; jlozic@unin.hr
2   Department of Logistics and Sustainable Mobility, University North, Trg dr. Žarka Dolinara 1, 48000 Koprivnica, Croatia; ivana.martincevic@unin.hr
*   Correspondence: kcikovic@unin.hr

**Abstract:** The supplier selection process is a strategic decision-making process that influences the company's sustainability. Lately, the sustainability concept has been highlighted as an organization's source of success and profitability. Therefore, the selection of a sustainable supplier has become an imperative for organizations and is the focus of this manuscript. Suppliers are key stakeholders in the supply chain, and their proper selection is a key factor in a successful and sustainable supply chain. For this reason, it is crucial to determine how and which methods are mostly used by companies when choosing sustainable suppliers with the aim of examining whether the Data Envelopment Analysis (DEA) contributes to the same. This article is the first to present a comprehensive bibliometric analysis of 87 articles dealing with the application of DEA in the sustainable supplier selection in the period 2010–2022, with the application of the keywords "Data Envelopment Analysis", "Supplier", and "Sustainable" in Scopus and Web of Science databases. The main goal of this manuscript is to explore the applications of DEA in a sustainable supplier selection and to provide an analysis and visualization of bibliometric data to reveal the annual trends of published articles in this area, the top contributing journals, the most cited papers, the most contributing authors, citations, affiliations, and countries' analysis, and an in-depth keyword visualization analysis. The findings of this study provide valuable insights and emphasize the ever-growing trend toward the selection of sustainable partners and suppliers in business using DEA methodology. Notably, this work shows the applicability and efficacy of DEA in specialized areas of supply chain management and should contribute to the construction of an overview of the existing literature on DEA studies regarding the process of selection of sustainable suppliers in supply chain management as well as stimulate the interest in the topic. This article gives an overview of a research field that is actually insufficiently explored through the scientific literature and presents a wide area and guidelines for future work.

**Keywords:** data envelopment analysis (DEA); efficiency measurement; non-parametric approach; sustainability; sustainable supplier selection; supply chain; bibliometric review

## 1. Introduction

Purchasing and supply chain management have a key role in creating a company's competitive edge. Moreover, purchasing represents "a strategic role in supply chain management for a firm and is the driver of competitive advantage" [1]. Therefore, the selection of suppliers in any business is one of the vital decision-making processes that have a major impact on the business' viability and sustainability. "Supplier selection is an important decision-making problem which involves many quantitative and qualitative factors incorporating vagueness and imprecision" [2]. The importance of a supplier selection whose actions affect the performance and productivity of the supply chain is a process that requires quality assessment, evaluation, and, ultimately, the selection of suppliers that will

contribute to an increase in the overall efficiency of the supply chain, reduced procurement costs, as well as an increase in market competitiveness [3].

However, the changes in the societies, as well as the increasing importance of corporative social responsibility, have imposed even greater importance on the selection and cooperation with sustainable, eco, and green suppliers.

The concept of sustainable development, which is connected to the selection of sustainable suppliers, emphasizes the need and necessity of the company to change the current business, values, rules, attitudes at all levels, activities, and business processes. Economic development and growth and the process of globalization and internationalization of society and markets cannot be stopped, but it is necessary to think in the direction of ensuring a healthy, sustainable, and green society, and a quality economy that will have less harmful effects on the environment. Therefore, it is no wonder that the selection of sustainable suppliers that ensures sustainable development is one of the key elements needed to formulate and implement development policies in the world. The key factor for a successful supply chain is the selection of sustainable suppliers but also the selection of the best sustainable strategy in the supply chain. Application of the DEA model in the process of selecting sustainable suppliers and the best sustainable strategy in the supply chain contributes to increasing the effectiveness and efficiency of the entire supply chain. It also contributes to the sustainability concept in the supply chain design [4].

This encapsulates the reason why DEA methodology has been of interest for researchers in studies regarding the process of selection of sustainable suppliers. DEA is the leading non-parametric approach for measuring the relative efficiency and benchmarking of peer decision-making units (DMUs). Ever since its introduction in 1978 by the seminal paper of Charnes, Cooper, and Rhodes [5,6], it has been widely recognized as one of the most effective methodologies for measuring efficiency and performance. Moreover, there are some specific areas in which DEA has been most applied [7]. More on the DEA methodology is laid out in Section 2.

To research and review novelties and benefits in the field of DEA and its application in the selection of sustainable suppliers, numerous pieces of world literature were researched. In this manuscript, an extensive bibliometric-based survey was conducted to summarize past findings in a research field—in this case, findings on DEA and its application in the selection of sustainable suppliers. Papers within the Scopus and Web of Science (WoS) databases were used to gather information on the research topic, which shows that the field of DEA has been researched by a large number of authors but is still insufficient in the field of its application in the selection of sustainable suppliers. This research and literature gap were the main motivation behind this research. The results of searching Scopus and Web of Science databases are presented in order to demonstrate the scarcity of this research topic, which resulted in 87 scientific papers dealing with DEA and its application in the selection of sustainable suppliers. The number of papers that investigate the issue of selection of sustainable suppliers with DEA methodology increased substantially from 2017 to 2022. Most of the research papers were from Iran, China, and the USA. In the research approach, the focus is only on papers in the field of DEA that are linked to the selection of sustainable suppliers.

Therefore, the main goal of the present work is to perform an extensive literature review with bibliometric analysis of studies integrating DEA in a sustainable supplier selection in an attempt to answer the main research question: what are the current avenues of research for such studies? The results obtained are further complemented with a thematic analysis of real-world applications. There are not many studies that explore the application of DEA in the selection of sustainable suppliers and its importance and contribution to strengthening sustainable supply chain management, so this research contributes to strengthening the awareness of organizations about its application. Application of DEA is relatively new in a process of selection of sustainable suppliers so it needs to be strongly and systematically developed further especially with its impact on development of sustainable supply chains. By analyzing the relevant scientific sources, it can be

concluded that a strategic approach to the application and use of DEA in the selection of sustainable suppliers can provide numerous advantages and benefits. Based on the review and analysis of previous world research and based on the analysis of the application of the DEA methodology, several additional research questions are asked: (1) How does the use of the DEA methodology by companies affect the supply chain, and is this presented through researched literature? (2) Does the application of the DEA methodology by companies affect the quality and better selection of suppliers, i.e., sustainable suppliers, and is this presented through researched literature? (3) Is there a connection between the application of the DEA methodology when companies choose suppliers (they do not care about sustainability) and when they choose sustainable suppliers, and is this presented through researched literature? These questions are posed as additional research questions to the main research question to find out whether the DEA methodology affects and is a useful tool to select sustainable suppliers recognized by companies that can ultimately affect the sustainable supply chain presented through bibliographic review of the literature. Moreover, this study reveals the annual trends of published articles in this area, the top contributing journals, the most cited papers, the most contributing authors, citations, affiliations and countries' analysis, and an in-depth keyword visualization analysis.

The main contributions to this bibliometric review can be summarized as follows: (a) A bibliometric-based survey on the DEA applications for selection of sustainable suppliers is conducted; (b) Clarivate WoS and Scopus databases are explored for collecting the relevant studies; (c) Interesting statistical information from the DEA literature regarding DEA applications in the process of sustainable suppliers' selection is extracted; (d) The main real-world applications and case studies of DEA are categorized; (e) The main research directions and gaps are suggested for future studies that employ DEA in the sustainable supplier selection process.

The remainder of this paper is structured as follows: In Section 2, a theoretical framework regarding the sustainable suppliers and the DEA methodology is given. Section 3 presents the results from DEA articles in the selection of sustainable suppliers, including the research approach, publication years, document types, keywords analysis, the authors and journals analysis, affiliations analysis, citations analysis, and the co-authorship analysis based on countries. Section 4 opens up a discussion and presents the implications of this research and introduces the future trends for DEA in the selection of sustainable suppliers. The last, fifth section concludes the findings from the bibliometric literature review.

## 2. Theoretical Framework

### 2.1. Sustainable/Green Suppliers

Supply chains enable and represent the flow of goods, services, and information without which it is impossible to imagine the normal functioning of the market. They include and are part of the primary, secondary, and tertiary sectors and cover the entire process and flow of production [8]. Supply chains create added value in the business process, which is why it is necessary to manage the supply chain in order to create added value constantly. The supply chain management process includes an "integrated process that includes planning and managing all stakeholder selection activities, procurement of materials, transformation of materials into the final product, as well as related logistics activities within the entire chain" [9] (p. 187). In order to examine the efficiency of the supply chain, the "Game-theoretical Design Technique" approach was tested. The design approach allows easier access to business decisions within the supply chain when the supply chain is exposed to some uncertain parameters and ensures agile, cooperative, and resource-efficient design of multi-stage supply chains [10].

The supply chain is evolving rapidly, and while this development follows the technology development, it also intends to ensure sustainable development. Many authors equate sustainable development with economic and social growth. Sustainable development is the aspiration of society to achieve sustainable economic growth to the extent that it will meet the needs of present and future generations. It is necessary to preserve production capacity

for the long term while achieving social goals such as increasing real income per capita, improving hygiene and nutrition, educational achievement, access to resources, equitable distribution of wealth, and increasing freedom [11–14]. The key goal of any organization is to ensure sustainability and create sustainable business models. Such an approach and business model rely primarily on the selection of sustainable and green suppliers that will ensure the path to sustainable development. Supply chains and their management and governance are a complex system that strives to minimize costs and maximize the level of service, and its focus today should primarily be on the selection of sustainable (green) suppliers.

Sustainable (green) suppliers improve and assist sustainable business models in the field of supply chain management. With an emphasis on sustainability and environmental care, the selection of green suppliers should be a central component and goal in supply chain management [15]. Suppliers and supplier relationship management create several benefits for the company—from creating long-term and loyal partners to creating greater visibility through better communication with them. Choosing a reliable business partner is the task of the management of companies that want to achieve long-term competitiveness and sustainability in the market. Sustainable (green) supplier selection and assessment are the most significant and complex challenges for supply chain managers [16,17]. Supplier selection is "intrinsically related to the Multi-Criteria Decision Making (MCDM) problem" [18]. Moreover, the process of selecting a supplier represents a "key competence in the sourcing function" [19]. Today, everything is intensified through environmental (green) factors that have, i.e., should have an impact on the selection of reliable and green suppliers. The question of how to identify and select sustainable suppliers can be answered through a system of weights that determine environmental factors as important decision factors when selecting suppliers by changing the DEA methodology [20]. One of the problems that arises in the business environment today is the selection of suppliers, primarily the selection of sustainable suppliers. Quality and proper selection of suppliers enable and facilitate the cross-efficiency fuzzy DEA technique. This method evaluates the efficiency of the supply chain and all its members (primarily in this case—suppliers) with the correct ranking which then allows the correct selection of suppliers and later reflects on the overall efficiency, costs, and delivery time within the supply chain [21].

Choosing sustainable suppliers means choosing business partners with the most beneficial monetary value on the one hand and the least harmful impact on society and the environment on the other [22]. Assessing the sustainability and performance of the supplier and the supply chain itself requires a selection of optimal tools and methods for the selection process, where DEA is an important tool for measuring the performance of sustainable suppliers and sustainable supply chains [23]. That the DEA methodology and the evaluation index method are an objective and quantitative tool for evaluating and selecting sustainable (green) suppliers is a fact confirmed by numerous researchers through their empirical research [23–30].

However, considering the importance of suppliers in the strategy framework of supply chains, it is rather surprising that "the sourcing function has not been subject to more focused research on the development of adequate decision support tools" [19].

### 2.2. Data Envelopment Analysis (DEA)

The DEA is a linear mathematical programming technique that is used for the evaluation of the performance (i.e., the relative efficiencies) of a group of complex entities referred to as Decision-Making Units (DMUs) [31]. DEA is one of the most widely applied non-parametric methodologies ever since its introduction in 1978 by Charnes, Cooper, and Rhodes (1978) [5,6] that has grown into a powerful mathematical and linear programming technique. Namely, according to [7], there are five areas in which DEA has been most applied, and these are Agriculture, Banking, Supply Chain, Transportation, and Public Policy. Research conducted in the field of supply chain related to planning and resource management in intermodal terminals proves that the application of the DEA model allows

planning decisions under conflicting requirements (the DEA method provides data on the number, capacity, and allocation of resources to address increasing flows) [32]. Moreover, the positive implication and real case study regarding the usage of the DEA method in the supply chain is used "to determine the efficiency of the rescheduled timetable (in terms of reduction of delays and maximization of robustness) and to rank alternatives according to their efficiency values [33] (p. 256).

DEA is a non-statistical and non-parametric approach that "makes no assumptions regarding the distribution of inefficiencies or the functional form of the production function (although it does impose some technical restrictions such as monotonicity and convexity)" [34], which represents one of its main advantages over parametric methodologies. It is a "data-oriented" method that converts multiple inputs to multiple outputs when evaluating peer units—DMUs [35]. DEA focuses on the extreme observations, which is its main distinction from the parametric methodologies that "focus on average tendencies and deviations from it". Moreover, DEA can employ multiple inputs and outputs, while parametric methodologies can only employ one output, which represents one of their biggest limitations [36]. This makes DEA "an excellent data-oriented efficiency analysis method" when using multiple inputs and outputs and "a useful performance evaluation and decision-making tool" [37].

DEA identifies the relative efficient DMUs in the observed sample that shape the efficiency frontier that measures the inefficiency of inputs or outputs of the other DMUs in the sample by comparing them and benchmarking with the relative efficient DMUs. Thus, DEA is also an econometric frontier method. The DMUs that lie on the frontier have the best relative efficiency, whereas the ones that are inefficient lie below the efficiency frontier [38]. Moreover, the results from the DEA vary from 0 to 1 (or 0 to 100%), with 1 being relatively efficient and a result below 1 being relatively inefficient. Thus, this approach enables a simple comparison of the DMUs in the sample [39,40].

Two basic DEA models are named in honor of their founders (CCR—Charnes, Cooper, and Rhodes and BCC—Banker, Charnes, and Cooper [5,6]). The CCR model employs a constant return to scale (CRS) assumption, i.e., "the output variables increase proportionally with input variables" [41], whereas the BCC DEA model employs a variable returns-to-scale (VRS) assumption, assuming that the proportional change in inputs does not necessarily lead to a proportional change in the outputs. The CCR model is graphically represented as a straight line, whereas the BCC model is represented by a convex hull [42].

However, it should be kept in mind that DEA is a methodology that explores the relative (and not absolute) efficiency within the analyzed sample of peer decision-making units [43]. The limitations and downsides of DEA are often scholarly researched, but most of the authors agree that "its advantages outweigh its limitations" and DEA should be considered "a significant diagnostic tool" [31].

In this bibliographic review, the applications of DEA in the process of supplier selection were taken into consideration. Moreover, this paper could represent an incentive to other scholars and researchers to employ the DEA methodology in the decision-making processes in any industry due to its advantages, easy implementation, and insights it provides.

## 3. Results on DEA Articles in the Selection of Sustainable Suppliers

### 3.1. Research Approach—A Survey and Bibliometric Analysis of DEA Literature Regarding the Selection of Sustainable Suppliers

In this research, the bibliometric analysis was conducted as a way of summarizing past findings in a research field (in this case, findings on the selection of sustainable suppliers using DEA). A systematic and comprehensive search was carried out in different databases, including the Web of Science (WoS) and Scopus databases. Relevant databases for this research were identified, and the focus was on the peer-review journals that are cited in Scopus and WoS (SSCI and SCI papers). Tables 1 and 2 present search strategies in WoS (SSCI and SCI) and Scopus (1990–2022 for the WoS database and 1980–2022 for the

Scopus database). In the mentioned periods, the papers appear by searching the mentioned databases according to the search strategy.

**Table 1.** WoS (SSCI, SCI) search strategy.

| Search Strategy | Hits | Timespan | Indexes |
|---|---|---|---|
| Data Envelopment Analysis (Title) AND Data Envelopment Analysis (Abstract), Data Envelopment Analysis (Keywords) | 2.371 | 1990–March 2022 | SCIEXPAND., SSCI, A&HCI, ESCI |
| Refined by: Supplier AND Sustainable | 18 | 2017–March 2022 | SCIEXPAND., SSCI, A&HCI, ESCI |

Source: Authors' work, 2022.

**Table 2.** Scopus search strategy.

| Search Strategy | Hits | Timespan | Indexes |
|---|---|---|---|
| TITLE-ABS-KEY (data AND envelopment AND analysis) | 4.991 | 1980–March 2022 | Scopus |
| Refined by: Supplier AND Sustainable | 795 | 2003–March 2022 | Scopus |

Source: Authors' work, 2022.

Through the first part of the search, WoS and Scopus were checked using keywords: "Data Envelopment Analysis" + "Supplier" + "Sustainable". This approach resulted in 813 hits (795 in Scopus and 18 in WoS). The aforementioned keywords are searched in "Title, Abstract, and Keywords" of documents belonging to Scopus and WoS databases.

The selection process of the papers for the bibliometric analysis is presented in Figure 1. According to Figure 1, the keywords "Data Envelopment Analysis + Supplier + Sustainable" were searched in "Title, Abstract, and Keywords" of documents belonging to WoS and Scopus databases. However, after merging all papers, 14 papers that were found in both databases were excluded. Therefore, 799 papers remained for the analysis. After reviewing the abstracts and keywords of all 799 papers, the papers that did not report the description of Data Envelopment Analysis + Supplier + Sustainable were eliminated. The paper was considered relevant if it fit the criterion of specifically covering the area of using DEA in the selection of sustainable suppliers (Figure 1). Finally, 87 publications remained after applying this exclusion criterion, and these 87 represent the basis for further analysis, including journal papers (JP), conference papers (CP), and review (RW) found in the DEA literature regarding the selection of sustainable suppliers using the DEA approach (2003–2022 regarding the Scopus database and 2017–2022 regarding the WoS database according to the search strategy).

*3.2. Publication Years, Document Types, and Keywords Analysis*

In the first step of the bibliometric analysis, the DEA literature regarding the selection of sustainable suppliers using the DEA approach is analyzed based on publication frequency during time horizon, document types, and the number of authors. The statistics based on basic information from the DEA literature regarding the selection of sustainable suppliers using the DEA approach are presented in Figure 2a–c. As shown in Figure 2a–c, the popularity of the DEA literature regarding the selection of sustainable suppliers using the DEA approach has increased, and a growing trend is apparent from 2017 until now, but it is also expected for the DEA approach in the field of selection of sustainable suppliers to grow even more in the future. This is understandable considering the overall increasing trend and awareness regarding sustainability in business and its positive impact as "a source of success, innovation, and profitability of companies" [44] throughout the world.

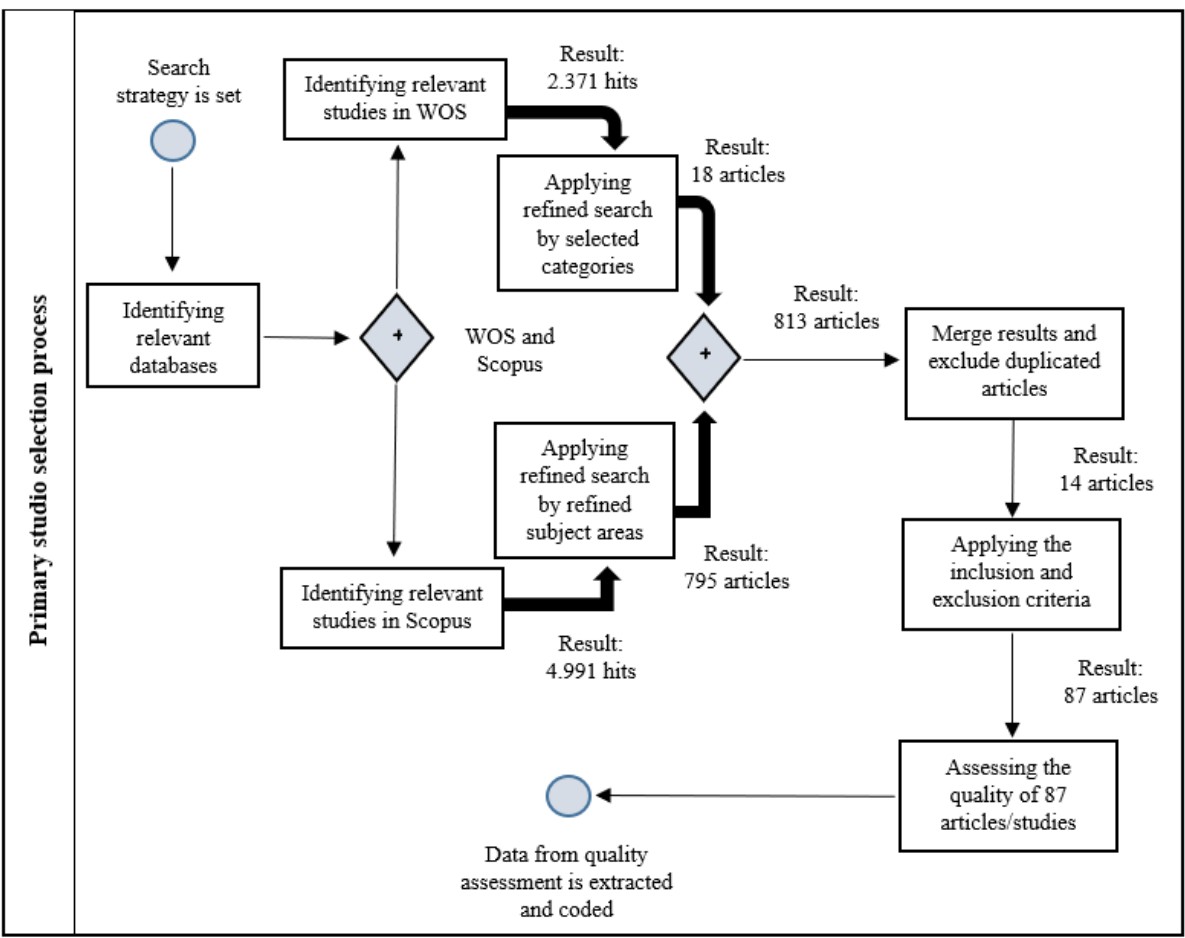

**Figure 1.** The selection process of the papers for the bibliometric analysis. Source: Author's work, 2022.

Sustainable development as well as sustainable management are global goals which require a long-term strategy of policy coordination for economic, social, and environmentally sustainable development. Today's businesses should strive to ensure and harmonize their own economic development and progress on the one hand and protect the environment on the other. The concept of sustainable development originated and began being mentioned in the 1980s, and the focus itself is on the connection between the development of society and environmental protection. Respecting the concept of sustainable development and doing business sustainably require companies to define strategies and create an economy that can regenerate itself and be sustainable. In order to ensure sustainable development in the company's business, numerous business transformations—from digital to technological—are necessary. This should all be conducted while respecting economic concepts which hold great potential and are possible solutions for sustainable development. Today, the strategy of sustainable development should be an integral part of the corporate strategy of the company, creating a business policy of the company aimed at the system of sustainable development. This way it contributes to the balance of company management through the cooperation of dynamic business and economic activities of the set system and changes caused by a dynamic ecological system.

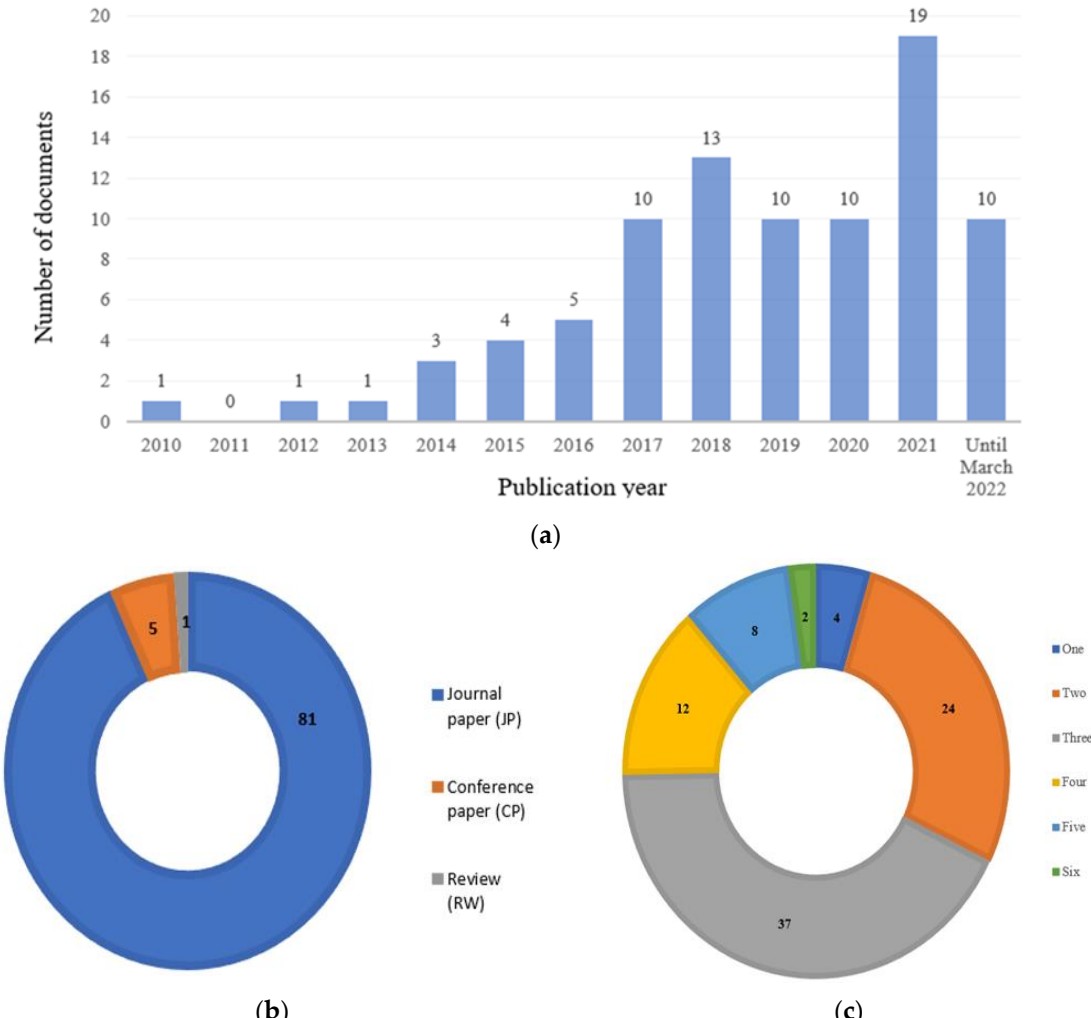

**Figure 2.** Basic bibliometric information of sustainable suppliers with DEA literature: (**a**) distribution of documents per year, (**b**) distribution of publication type, (**c**) distribution of the number of document authors. Source: Authors' work, 2022.

The word cloud of the DEA literature regarding the selection of sustainable suppliers using the DEA approach based on the most popular keywords in DEA is presented in Figure 3. A word cloud is "a visualization technique that provides the user with an overview of the content of a collection of texts", where the word size in the word cloud represents the frequency with which the words appear in the text collection [45]. Moreover, a "word cloud is a visual representation of word frequency", which means that the larger the words, the more frequently does the term appear within the text collection [46]. Word clouds have been widely used in bibliometric analysis as a guidance tool that helps identify the focus of the large set of published articles. The word cloud in this article was presented with the text mining technique and the word cloud option in R (Figure 3). This simple analysis and the tool we used prove that the search strategy of this article was chosen correctly, as the most popular keywords and key phrases which appear in the word cloud of the selection of sustainable suppliers with DEA also appear in the search keywords.

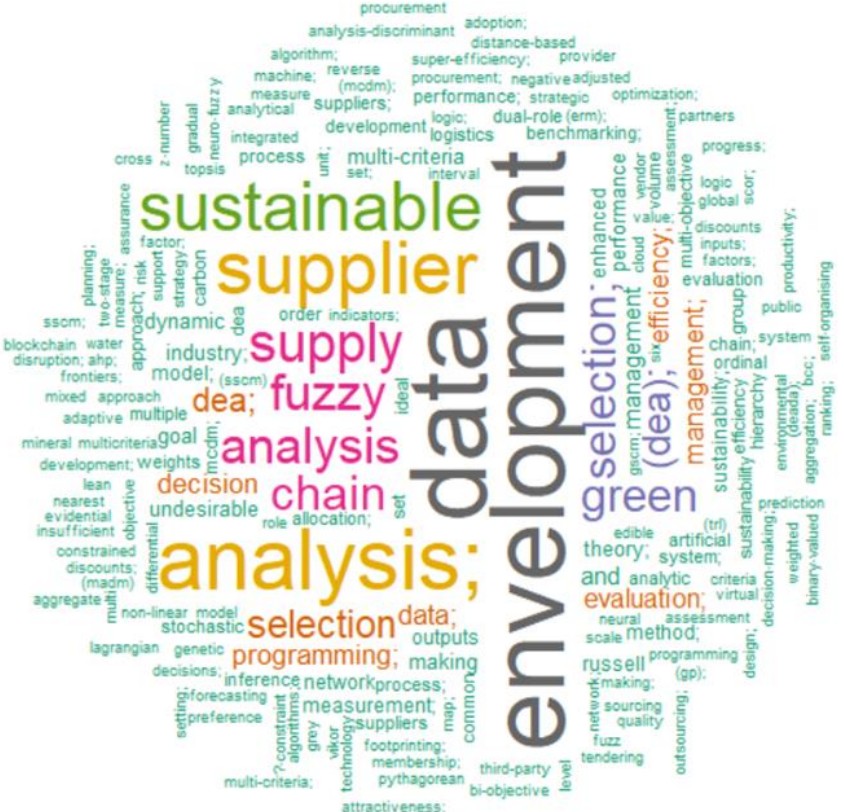

**Figure 3.** Word cloud of the selection of sustainable suppliers with DEA literature. Source: Authors' work, 2022.

*3.3. Authors and Journals Analysis*

In the second step of the bibliometric analysis, statistics based on authors and journals in the DEA literature regarding the selection of sustainable suppliers using the DEA approach are extracted. The top contributing authors and the most influential journals in the DEA field regarding the selection of sustainable suppliers using the DEA approach are proposed in Tables 3 and 4, respectively.

**Table 3.** The top contributing authors in sustainable supplier selection with DEA literature.

| Author | JP | CP | First | Second | Third | Fourth | Total |
|---|---|---|---|---|---|---|---|
| Saen R.F. | 18 | | | 11 | 5 | 2 | 18 |
| Izadikhah M. | 7 | | 6 | 1 | | | 7 |
| Yousefi S. | 6 | | 1 | 2 | 3 | | 6 |
| Azadi M. | 5 | | 3 | 1 | | 1 | 5 |
| Shabanpour H. | 4 | | 3 | 1 | | | 4 |
| Wang C.-N | 4 | | 4 | | | | 4 |
| Amindoust A. | 3 | 1 | 4 | | | | 4 |
| Dania W.A.P. | 3 | 1 | 2 | 2 | | | 4 |
| Dobos I. | 3 | 1 | 1 | | | | 2 |
| Vörösmarty G. | 3 | 1 | 2 | 2 | | | 4 |
| Tavana M | 3 | | 2 | | 1 | | 3 |
| Ahmadi K. | 3 | | | | 3 | | 3 |
| Jain V. | 3 | | 1 | 2 | | | 3 |
| Kumar S. | 3 | | | 1 | 2 | | 3 |
| Kumar A | 3 | | 2 | | 1 | | 3 |
| Li F. | 3 | | 2 | | | 1 | 3 |

**Table 3.** *Cont.*

| Author | JP | CP | First | Second | Third | Fourth | Total |
|---|---|---|---|---|---|---|---|
| Moheb-Alizadeh H | 3 | | 3 | | | | 3 |
| Handfield R. | 3 | | | 3 | | | 3 |
| Jauhar S.K. | 2 | 1 | 3 | | | | 3 |
| Jafarzadeh Ghoushchi S | 2 | | 1 | 1 | | | 2 |
| Chandra C. | 2 | | | | | 2 | 2 |
| Pant M. | 2 | | | 2 | | | 2 |
| Rashidi K. | 2 | | 2 | | | | 2 |
| Raut R. | 2 | | 1 | 1 | | | 2 |
| Kharat M. | 2 | | | 1 | | 1 | 2 |
| Sharafi H. | 2 | | 1 | 1 | | | 2 |
| Soltanifar M | 2 | | 1 | 1 | | | 2 |
| Zhang Z | 2 | | 1 | | | 1 | 2 |
| Zhou X | 2 | | 2 | | | | 2 |

Source: Authors' work, 2022.

**Table 4.** The top journals contributing to the sustainable supplier selection with DEA literature.

| Journal | 2010–2015 | 2016 | 2017 | 2018 | 2019 | 2020 | 2021 | 2022 | Total |
|---|---|---|---|---|---|---|---|---|---|
| International Journal of Production Economics | 2 | | | 1 | 1 | | 1 | 1 | 6 |
| Journal of Cleaner Production | 1 | | 2 | | 1 | 1 | 1 | | 6 |
| Sustainability (Switzerland) | | | 2 | | | | 2 | 2 | 6 |
| Computers and Industrial Engineering | | | | 1 | 2 | | 1 | | 4 |
| Annals of Operations Research | | | | 1 | | 1 | | 1 | 3 |
| Benchmarking | | | | 1 | | 1 | | | 2 |
| Computers and Operations Research | 1 | | | 1 | | | | | 2 |
| Environmental Science and Pollution Research | | | | | | | 1 | 1 | 2 |
| Group Decision and Negotiation | | | | | 1 | | | 1 | 2 |
| Industrial Management and Data | | | | | | | 1 | 1 | 2 |
| International Journal of Industrial and Systems Engineering | | | 1 | 1 | | | | | 2 |
| Neural Computing and Applications | | 1 | 1 | | | | | | 2 |
| Transportation Research Part D: Transport and Environment | | | 2 | | | | | | 2 |

Source: Authors' work, 2022.

The identification of the top contributing authors presents the authors who have the highest impact in the research field. In addition, the presentation of the top journals helps researchers to identify journals that are suitable to submit their papers. In Table 3, the number of research works by each author is presented separately by the type of publication (JP, CP) and the order of the author (1st, 2nd, 3rd, 4th) among the authors of the research.

As seen in Table 3, Saen R.F. from the Department of Industrial Management, Faculty of Management and Accounting, Karaj Branch, Islamic Azad University, with 18 co-authored research works, dominates the list of top contributing authors in the DEA literature regarding the selection of sustainable suppliers using the DEA approach. Izadikhah M. from the

Department of Mathematics, College of Science, Arak Branch, Islamic Azad University with 7 research works and one co-authored paper is also a top contributing author. The authors' analysis of the DEA field regarding the selection of sustainable (green) suppliers using the DEA approach from a qualitative aspect shows that authors Kuo R.J., Wang Y.C., and Tien F.C. are the pioneering researchers in the DEA field regarding the selection of sustainable (green) suppliers using the DEA approach and are the top-cited authors in this field.

According to Table 4, articles in the DEA field regarding the selection of sustainable (green) suppliers using the DEA approach were published in top-tier journals such as Annals of Operations Research (3 articles), Benchmarking (2 articles), Computers and Industrial Engineering (4 articles), Computers and Operations Research (2 articles), Environmental Science and Pollution Research (2 articles), Group Decision and Negotiation (2 articles), Industrial Management and Data (2 articles), International Journal of Industrial and Systems Engineering (2 articles), International Journal of Production Economics (6 articles), Journal of Cleaner Production (6 articles), Neural Computing and Applications (2 articles), Sustainability (Switzerland) (6 articles), and Transportation Research Part D: Transport and Environment (2 articles). This indicates the attractiveness of the DEA approach regarding the selection of sustainable (green) suppliers using the DEA methodology.

### 3.4. Affiliations Analysis

As a third step of the bibliometric analysis, a statistical overview of the top contributing affiliations in sustainable supplier selection with the application of DEA methodology was conducted and is presented in Table 5. A brief presentation of the most influential countries/territories as well as organizations/institutions along with the number of publications containing the DEA literature regarding the selection of sustainable suppliers using the DEA approach is laid out. According to the results shown in Table 5, the top 10 contributing countries in the DEA area regarding the selection of sustainable suppliers using the DEA approach are Iran, China, the United States, India, Taiwan, Hungary, Malaysia, Australia, Germany, and Canada. On the other hand, the top five contributing universities and institutions in the DEA field regarding the selection of sustainable suppliers using the DEA approach are Islamic Azad University, North Carolina State University, Indian Institute of Technology, Corvinus University of Budapest, and the National Institute of Industrial Engineering.

**Table 5.** Top contributing affiliations in the sustainable supplier selection with DEA literature.

| Countries/Territories | Number of Publications | Organizations/Institutions | Number of Publications |
|---|---|---|---|
| Iran | 60 | Islamic Azad University | 27 |
| China | 28 | North Carolina State University | 6 |
| United States | 20 | Indian Institute of Technology | 5 |
| India | 18 | Corvinus University of Budapest | 4 |
| Taiwan | 15 | National Institute of Industrial Engineering | 5 |
| Hungary | 7 | Sohar University | 3 |
| Malaysia | 7 | La Salle University | 3 |
| Australia | 6 | University of Malaya | 3 |
| Germany | 5 | La Salle University, Philadelphia | 3 |
| Canada | 5 | University of St. Thomas | 3 |
| Oman | 4 | Budapest University of Technology and Economics | 3 |
| Slovenia | 3 | University of Chinese Academy of Sciences | 3 |
| United Arab Emirates | 3 | Universitas Brawijaya, Malang, Indonesia | 2 |
| United Kingdom | 3 | University of Technology, Sydney | 2 |
| Indonesia | 2 | National Kaohsiung University of Applied Sciences | 3 |

**Table 5.** *Cont.*

| Countries/Territories | Number of Publications | Organizations/Institutions | Number of Publications |
|---|---|---|---|
| South Africa | 2 | University of Tehran | 3 |
| Sweden | 2 | University of Paderborn | 3 |
| Vietnam | 2 | Fuzhou University | 2 |
| Belgium | 2 | University of British Columbia | 2 |
| Denmark | 1 | National Institute of Technology | 3 |
| France | 1 | University of Sharjah | 3 |
| Japan | 1 | University of Gothenburg | 2 |
| Mexico | 1 | Fortune Institute of Technology | 2 |
| Singapore | 1 | Griffith Business School, Griffith University | 2 |
| Norway | 1 | National Kaohsiung University of Applied Sciences | 2 |
| New Zealand | 1 | University of Technology, Sydney | 2 |
| Saudi Arabia | 1 | Dongbei University of Finance and Economics | 2 |
| Switzerland | 1 | University of Ljubljana | 2 |
| Poland | 1 | University of Michigan–Dearborn | 2 |
| Tunisia | 1 | Indian Institute of Management | 2 |
| Russian Federation | 1 | Urmia University of Technology | 2 |

Source: Authors' work, 2022.

### 3.5. Citations Analysis

Out of the total number of published articles implementing DEA in the sustainable supplier selection (87), the number of cited articles is 73 (83.91%). In the fourth phase of this bibliometric analysis, the statistics regarding citations of the surveyed papers were analyzed based on data from the Scopus and WoS databases.

The list of the most cited research in the field of sustainable supplier selection is presented in Table 6. As seen in Table 6, the most cited documents in the research field of sustainable supplier selection with DEA were published in top-tier journals such as the International Journal of Production Economics (4 articles), Journal of Cleaner Production (4 articles), Annals of Operations Research (3 papers), Computers and Industrial Engineering (3 articles), Environmental Science and Pollution Research (2 papers), Computers and Operations Research (2 articles), Group Decision and Negotiation (2 articles), Neural Computing and Applications (2 articles), Sustainability (2 articles), Transportation Research Part D: Transport and Environment (2 articles), Benchmarking (1 article), Omega (1 article), Processes (1 article), Mathematics (1 article), Advanced Materials Research (1 article), and Applied Soft Computing Journal (1 article). The citation analysis implies the importance of the selection of sustainable suppliers.

**Table 6.** Top cited documents in sustainable supplier selection with DEA literature.

| Year | Author (s) | Publication | Citations |
|---|---|---|---|
| 2010 | Kuo R.J., Wang Y.C., Tien F.C. | Journal of Cleaner Production | 443 |
| 2015 | Azadi M., Jafarian M., Saen R.F., Mirhedayatian S.M. | Computers and Operations Research | 247 |
| 2014 | Kumar A., Jain V., Kumar S. | Omega (United Kingdom) | 199 |
| 2014 | Bai C., Sarkis J. | Supply Chain Management | 143 |
| 2019 | Rashidi K., Cullinane K. | Expert Systems with Applications | 114 |
| 2014 | Dobos I., Vörösmarty G. | International Journal of Production Economics | 109 |
| 2015 | Mahdiloo M., Saen R.F., Lee K.-H. | International Journal of Production Economics | 93 |

**Table 6.** *Cont.*

| Year | Author (s) | Publication | Citations |
|------|-----------|-------------|-----------|
| 2016 | Fallahpour A., Olugu E.U., Musa S.N., Khezrimotlagh D., Wong K.Y. | Neural Computing and Applications | 91 |
| 2016 | Zhou X., Pedrycz W., Kuang Y., Zhang Z. | Applied Soft Computing Journal | 75 |
| 2017 | Hatami-Marbini A., Agrell P.J., Tavana M., Khoshnevis P. | Journal of Cleaner Production | 69 |
| 2017 | Shabanpour H., Yousefi S., Saen R.F. | Journal of Cleaner Production | 59 |
| 2019 | Moheb-Alizadeh H., Handfield R. | Computers and Industrial Engineering | 49 |
| 2015 | Shi P., Yan B., Shi S., Ke C. | Information Technology and Management | 46 |
| 2018 | Amindoust A. | Computers and Industrial Engineering | 40 |
| 2019 | Pishchulov G., Trautrims A., Chesney T., Gold S., Schwab L. | International Journal of Production Economics | 34 |
| 2018 | Moheb-Alizadeh H., Handfield R. | International Journal of Production Research | 33 |
| 2017 | Shabanpour H., Yousefi S., Farzipoor Saen R. | Transportation Research Part D: Transport and Environment | 32 |
| 2016 | Jain V., Kumar S., Kumar A., Chandra C. | Journal of Manufacturing Systems | 30 |
| 2017 | Tavana M., Shabanpour H., Yousefi S., Farzipoor Saen R. | Neural Computing and Applications | 29 |
| 2019 | Cheaitou A., Larbi R., Al Housani B. | Socio-Economic Planning Sciences | 29 |
| 2017 | Izadikhah M., Farzipoor Saen R., Ahmadi K. | Transportation Research Part D: Transport and Environment | 28 |
| 2018 | Wang C.-N., Nguyen V.T., Thai H.T.N., Tran N.N., Tran T.L.A. | Mathematics | 28 |
| 2018 | Raut R., Kharat M., Kamble S., Kumar C.S. | Benchmarking | 28 |
| 2018 | Jafarzadeh Ghoushchi S., Dodkanloi Milan M., Jahangoshai Rezaee M. | Journal of Industrial Engineering International | 27 |
| 2018 | Zarbakhshnia N., Jaghdani T.J. | International Journal of Advanced Manufacturing Technology | 25 |
| 2021 | Kaur H., Prakash Singh S. | International Journal of Production Economics | 25 |
| 2020 | Tavassoli M., Saen R.F., Zanjirani D.M. | Sustainable Production and Consumption | 22 |
| 2019 | Torres-Ruiz A., Ravindran A.R. | Computers and Industrial Engineering | 20 |
| 2019 | Wu M.-Q., Zhang C.-H., Liu X.-N., Fan J.-P. | IEEE Access | 20 |
| 2018 | Boudaghi E., Farzipoor Saen R. | Computers and Operations Research | 18 |
| 2017 | Izadikhah M., Saen R.F., Ahmadi K. | Asia-Pacific Journal of Operational Research | 17 |
| 2020 | Wang C.-N., Tsai H.-T., Ho T.-P., Nguyen V.-T., Huang Y.-F. | Processes | 17 |
| 2016 | Kumar A., Jain V., Kumar S., Chandra C. | Enterprise Information Systems | 16 |
| 2017 | Wang C.-N., Ho H.T., Luo S.-H., Lin T.-F. | Sustainability (Switzerland) | 16 |

**Table 6.** *Cont.*

| Year | Author (s) | Publication | Citations |
|---|---|---|---|
| 2018 | Choudhury N., Raut R.D., Gardas B.B., Kharat M.G., Ichake S. | International Journal of Business Excellence | 16 |
| 2017 | Yu M.-C., Su M.-H. | Sustainability (Switzerland) | 14 |
| 2013 | Amindoust A., Ahmed S., Saghafinia A. | Advanced Materials Research | 12 |
| 2015 | Jauhar S.K., Pant M., Nagar M.C. | Computer Methods in Materials Science | 12 |
| 2018 | Izadikhah M., Saen R.F., Roostaee R. | Annals of Operations Research | 12 |
| 2020 | Vörösmarty G., Dobos I. | Journal of Cleaner Production | 10 |

Source: Authors' work, 2022.

Figure 4 shows the citations' overview and tracks citation data generated from a set of selected documents (87 selected documents) and introduces citation frequency in the field of sustainable supplier selection with the DEA literature with their number of documents as well as the h-index of sustainable supplier selection with the DEA literature. The h-index is based on the highest number of papers included that have had at least the same number of citations.

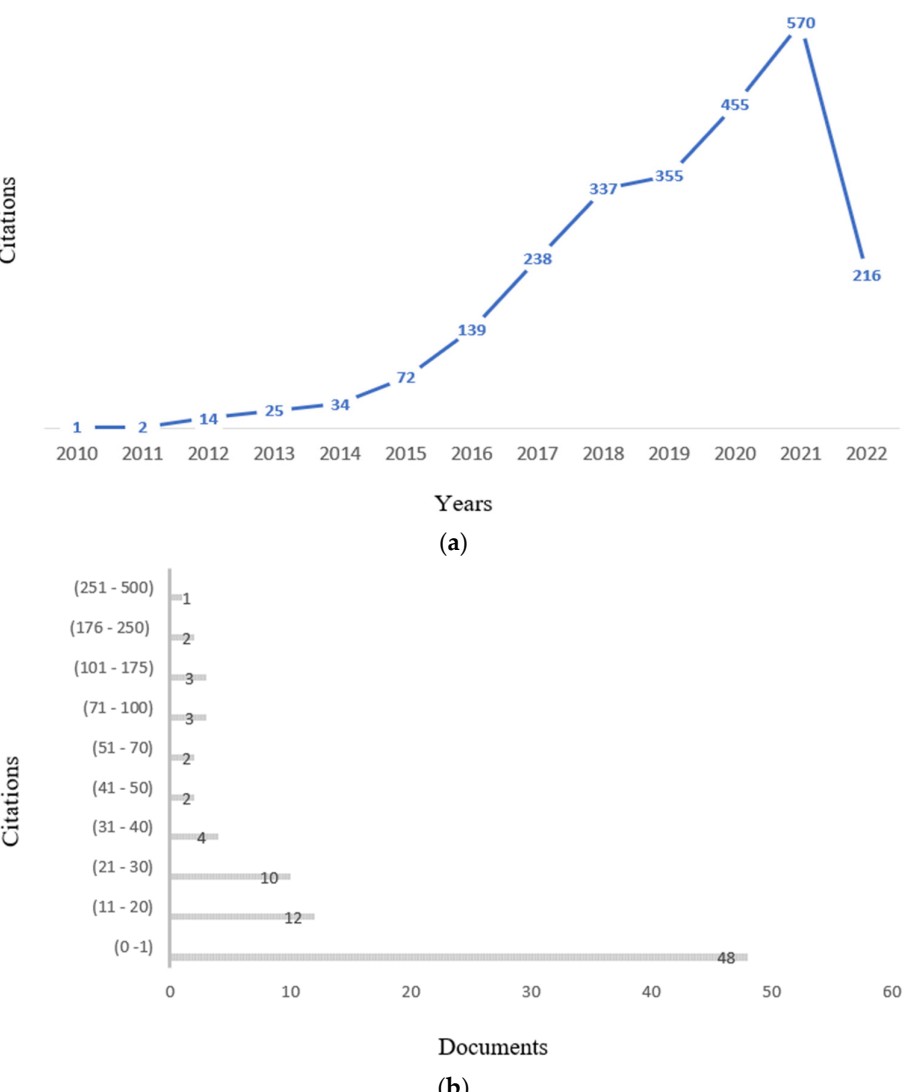

(a)

(b)

**Figure 4.** *Cont.*

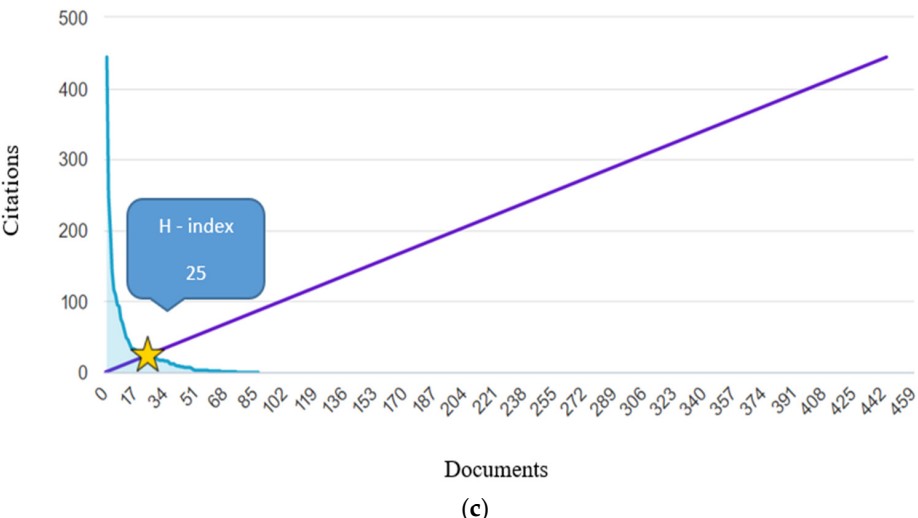

(**c**)

**Figure 4.** Citations analysis of sustainable supplier selection with DEA literature, (**a**) citation overview of selected documents in sustainable supplier selection with DEA literature, (**b**) frequency of citations in DEA literature regarding the sustainable supplier selection, (**c**) h-index of literature of sustainable supplier selection with DEA literature. Source: Authors' work, 2022.

The citation analysis enables researchers "to identify the most cited studies in a specific field and the evolution of their popularity over time" and it is believed that "the number of citations to a particular document indicates its impact on literature" [47].

### 3.6. Co-Authorship Analysis

With the help of the VOSviewer software, the network visualization for country co-authorship was created (Figure 5), and the network visualization of the top authors' density visualization is presented in Figure 6. A total of 32 countries were identified as contributing countries. However, only 7 of them are shown in Figure 5 since they are the most contributing countries in the area of sustainable supplier selection with DEA. Moreover, as shown in Figure 6, five authors appear in the density visualization due to their contribution to this field (these are Saen R.F., Izadikhah M., Yousefi S., and Azadi M.).

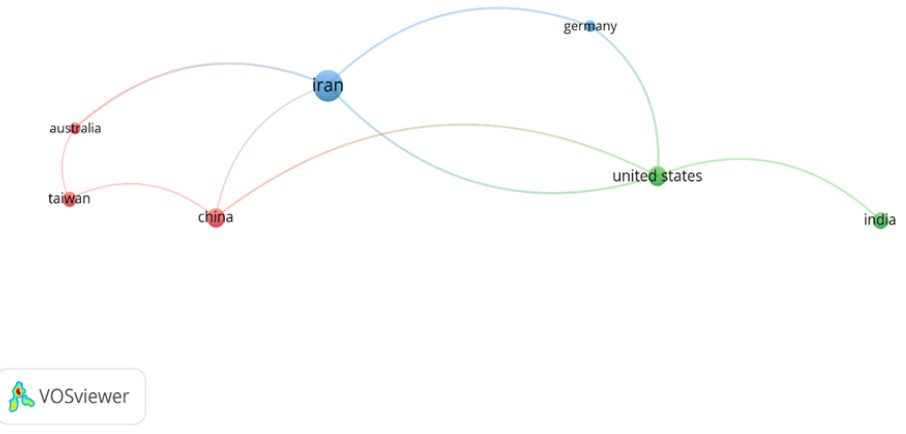

**Figure 5.** Network visualization by countries. Source: Authors' work, via VOSviewer software 2022.

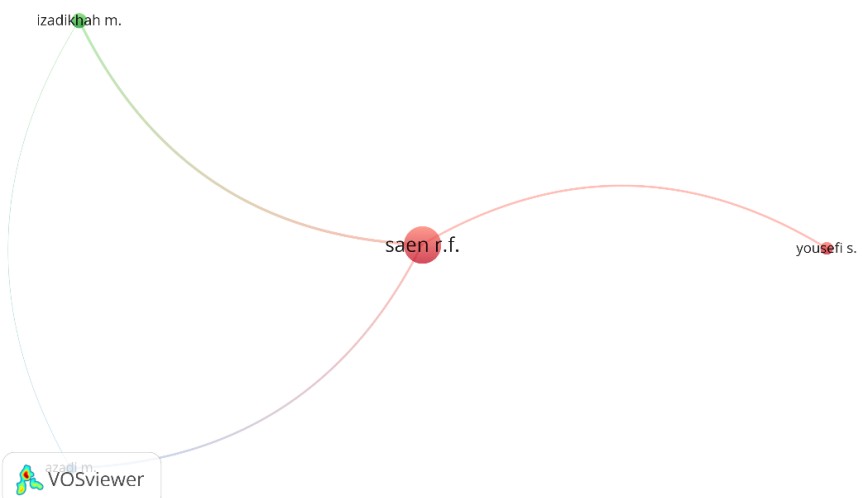

**Figure 6.** Network visualization of top authors' network. Source: Authors' work, via VOSviewer software 2022.

This analysis attempts to "shed light on clustered research among scholars from a particular region", and this information can be used "to justify and spark new research among scholars in underrepresented regions". Moreover, this visualization presents mapped collaborations between different countries and different authors and could inspire "prospective scholars to reach out to established and trending scholars in the respective research field" [48].

*3.7. Research Gap*

Regarding the research area, the obvious problem is the lack of existing research. By identifying relevant publications and cited literature in selected publications, keywords, abstract, and title, the DEA application in the selection of sustainable suppliers is showing a research gap. The DEA application in the selection of sustainable suppliers has not been studied and elaborated enough. It is required to extend the research in this area so it could be useful to academics but also managers in practice. It is necessary to encourage researchers to research in this area and to make the area interesting to as many researchers as possible in order to raise awareness of the importance of sustainable suppliers for the development of a sustainable supply chain. The key question is whether researchers and practitioners recognize and understand the DEA methodology, its application, and benefits so that it can be used in the selection of sustainable suppliers. There may be a problem of misunderstanding the DEA and how it is used in selecting sustainable suppliers, where the goal should be to make it recognizable and understandable to the wider community (both researchers and practitioners). It is important to address how the DEA exists since the 1980s and has proved useful in many areas, which is why it is necessary that the area related to the selection of sustainable suppliers will be also recognized through the DEA.

With all of the mentioned above, this review is relevant for academics but also has a practical relevance. It provides exploratory insights of the DEA benefits in the area of the selection of sustainable suppliers. This review can also make managers within the supply chain aware of choosing sustainable suppliers through the DEA methodology. This research can also assist business in setting sustainable supply chain strategies through the DEA by connecting with sustainable suppliers. This research should be viewed as an addition on the current way and process of how companies are selecting (sustainable) suppliers in adjusting their business strategy and thus affecting its supply chain. Within the process of selecting sustainable suppliers using DEA, the literature emphasized the importance of environmental factors as one of the main reasons where DEA can have the greatest impact in such selection. Therefore, the future role of DEA in the selection of sustainable suppliers should be further researched to determine how its influence will shift and what effects it

will have. Through presented research papers and its analysis, it is obvious that it can assist academics but also managers in practice in making and predicting how and which factors are crucial in changing behavior in the supply chain by selecting sustainable suppliers using DEA.

## 4. Discussion, Implications, and Future Trends for DEA in the Selection of Sustainable Suppliers

The DEA methodology is a powerful, popular mathematical programming technique that has found a wide application in many industries and decision-making processes. It is often combined with different MCDM techniques, which according to the survey were mostly techniques for order of preference by similarity to ideal solution (TOPSIS), analytic hierarchy process (AHP), Artificial neural network (ANN), analytic network process (ANP), fuzzy DEA, Green DEA (GDEA) models, etc. As opposed to the traditional approach to the supplier selection problem, "modern approaches are also considering the green aspects because of the rising importance of environmental impacts to comply with environmental standard" [1].

Additionally, there are some suggested guidelines and directions for further research in the sustainable supplier selection with the DEA area, which are as follows: there is an evident trend in developing the GDEA (green DEA) models for practical implications, and that is encouraged and expected from both scholars and managerial teams and future analysts; the AHP, TOPSIS, and ANN MCDM models in combination with DEA are increasingly popular in this area and "are already linked to strategic purchasing practices and relate to a more advanced level" [49]. The fuzzy DEA models have been increasingly implemented in this area (especially in the period from 2015 to 2022). As shown in Table 7, the most common research areas in which the DEA and MCDM models were combined are Artificial Intelligence, Supply Chain Management, Machine Learning, and Supplier Selection. However, there is huge potential for an even greater use of the combined DEA and MCDM models in many different industries in the future.

Thus, "to solve the MCDM problem in supplier selection (due to fact that sustainable supplier selection is here in focus) the study was conducted and it proposes a new fuzzy DEA model based on the super-efficiency DEA model, and confirms their interconnectedness" [50] (p. 91).

In addition, finally, as [47] put it, "the most valuable suggestion for future studies would be to apply existing or extended DEA approaches in real-world problems and applications".

**Table 7.** The combined use of DEA and MCDM models.

| Title of Paper | Keywords | Review and Discussion | Combination of DEA and MCDM |
|---|---|---|---|
| An integrated model for green supplier selection under fuzzy environment: application of data envelopment analysis and genetic programming approach [51] | Artificial intelligence; Data envelopment analysis (DEA); Ge-netic programming (GP); Green supplier selection; Parametric analysis | The paper research models based on hybrid artificial intelligence (AI) that deal with supplier evaluation. It highlights the artificial neural network (DEA-ANN) as one of the applied methods in the assessment of suppliers by a combination of artificial intelligence and DEA. The paper also investigates how to improve previous models and creates a new robust nonlinear mathematical equation for evaluating efficiency and selecting suppliers using established criteria and genetic programming. | DEA + Artificial Intelligence + Supplier Selection |

**Table 7.** *Cont.*

| Title of Paper | Keywords | Review and Discussion | Combination of DEA and MCDM |
|---|---|---|---|
| Sustainable supplier evaluation and selection with a novel two-stage DEA model in the presence of uncontrollable inputs and undesirable outputs: a plastic case study [28] | Multiple criteria decision-making; Supply chain management; Sustainable supplier selection; Two-stage data envelopment analysis; Uncontrollable inputs; Undesirable outputs | The study "proposes new two-stage DEA network model in the presence of uncontrolled inputs and undesirable outputs with consideration of a set of intermediates between the two phases for evaluation and selection of the best sustainable supplier" | DEA + Supplier Selection |
| Supplier selection considering sustainability measures: An application of weight restriction fuzzy-DEA approach [52] | Data envelopment analysis; Fuzzy set theory; Supplier selection; Sustainable development; $\alpha$-cut approach | The study proposes a new model for supplier evaluation and ranking and integrates fuzzy set theory and DEA into the new model, taking into account the decision-makers' preferences and resolving ambiguities and uncertainties in the supplier selection process. This paper presents and "developed a new fuzzy-DEA model, using the $\alpha$-cut approach and taking into account weight constraints". | DEA + Fuzzy-DEA model + Supplier Selection (automotive parts supplier) |
| A comparison of fuzzy DEA and fuzzy TOPSIS in sustainable supplier selection: Implications for sourcing strategy [53] | DEA; Logistics; Sourcing; Suppliers; Sustainability; TOPSIS | The paper combines two methods in supplier selection, "technique for ordering preference by similarity to ideal solution" (TOPSIS) and DEA. Based on a small number of evaluation criteria, the study proves that the combination of these two methods is applicable and useful for shortlisting potential sustainable suppliers (the paper suggestions for expanding research take the application of selected models to a number of criteria into account). | DEA + TOPSIS + Sustainable Supplier Selection |
| How to use fuzzy screening system and data envelopment analysis for clustering sustainable sup-pliers? A case study in Iran [54] | Data envelopment analysis (DEA); DEA-Based clustering method; Enhanced Russell model (ERM); Fuzzy screening system; Sustainable supply chain management | The study uses the DEA method to group suppliers into clusters and thus identifies and eliminates unqualified suppliers. The paper presents a new algorithm that uses the fuzzy screening system and the DEA method to select suppliers. | DEA + Fuzzy-DEA model + Supplier Selection |
| Production and scale efficiency of South African water utilities: The case of water boards [55] | Data Envelopment Analysis; Scale efficiency; Technical efficiency; Water boards; Water losses | The study applies the DEA model to be able to measure the technical efficiency of utility companies in South Africa. Therefore, the DEA serves to determine, measure, analyze, and compare the technical performance of all water panels in South Africa. | DEA + Supplier Selection |

**Table 7.** *Cont.*

| Title of Paper | Keywords | Review and Discussion | Combination of DEA and MCDM |
|---|---|---|---|
| A Hybrid Supplier Selection Approach Using Machine Learning and Data Envelopment Analysis [56] | Data Envelopment Analysis; Decision Tree; Kernel Support Vector Machine; Logistic Regression; Machine Learning; Supplier Selection | This study integrates and combines DEA and machine learning, where specific machine learning algorithms applicable to DEA results are presented. The paper focuses on developing a hybrid model for supplier selection by combining DEA methods and machine learning algorithms. | DEA + Supplier Selection + Machine Learning |

Source: Authors' work, 2022.

The DEA is expected to experience an even greater use by managers and data analysts in the process of decision-making. Moreover, managers ought to use the DEA as an efficiency-and-performance "supporting tool in the decision-making process" [57]. Having in mind the primary goal of the DEA, which is to provide a classification of relatively efficient and relatively inefficient units, this opens up vast opportunities for its wider use in supply chain management and specifically in the selection of sustainable suppliers. DEA has a wide usage in different industries, which proves its usefulness and benefits. As the research itself shows, DEA applications are experiencing growth from 2017 until now (2022), and researchers are realizing the necessity to research this area. However, the field is still insufficiently researched, and too few researchers deal with and analyze the advantages and benefits of DEA in selecting sustainable suppliers. Although the DEA methodology is not a new concept, the increase in its usage is evident in a number of different industries and different fields of business operations through the years, and its application changes how companies operate.

The results of the presented bibliometric literature review show that sustainable development and sustainable suppliers nowadays are imperative for doing business, and DEA influences and helps this process. Sustainable development and sustainable suppliers affect the business process and increase the efficiency and effectiveness of a company. A successful supply chain depends on quality suppliers and, above all else, sustainable suppliers, which affect the long-term growth and development of the company and make for greater visibility and respect for the environmental component of the company. The goal of the companies is to build a resilient supply chain while striving to strengthen all business processes and all business components of the company where sustainable suppliers are one of the necessary conditions that companies need to meet and fulfill. It is evident that according to the growing trend of the observed articles, this area is becoming the interest of an increasing number of researchers (although currently insufficient) and that the imperative of the company is sustainable growth and development, where sustainable suppliers are one of the key factors. In the field of scientific research, it is necessary to place increasing emphasis on the importance of sustainable suppliers and to encourage research in this area in order to highlight their importance in today's modern and global business. The idea and the aim of the paper was to present and show that sustainable suppliers are indispensable stakeholders in the supply chain that facilitate and contribute to the success of the entire process of the supply chain.

This study provides a summary of the work that was conducted to develop models and approaches for the selection of sustainable suppliers. Moreover, an overview of the real-life applications of DEA in the sustainable supplier selection was created in Figure 7. The process of selection of a sustainable supplier with DEA has been widely applied in various industries. According to Figure 7, DEA was extensively used in real-life decision-making situations, such as the food processing industry (oil and palm oil industry), the manufacturing industry, the textile and apparel industry, the semiconductor industry, the

automotive industry, the logistics industry, the payment and banking industry, the cellphone industry, the construction industry, the urban transportation industry, and even the civil society industry and the blockchain technology industry. This implies the applicability and effectiveness of DEA in the decision-making process in supply chain management.

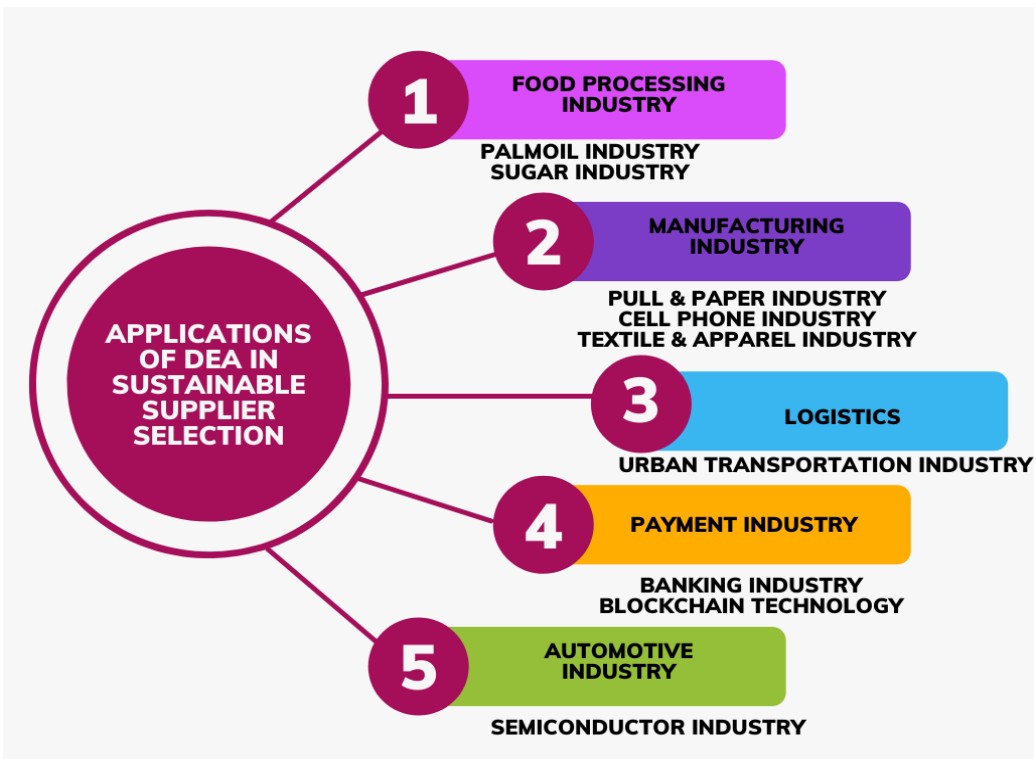

**Figure 7.** The main real-life applications and case studies of the DEA approach in sustainable supplier selection. Source: Authors' work, 2022.

Moreover, this research area needs more empirical, theoretical, and quantitative evidence and is the scope for many studies to be carried out in the future. The conducted research gives a brief overview of a research field that is insufficiently explored through the scientific literature and presents a wide area for further research.

## 5. Conclusions

This study was designed to survey articles that implement the DEA methodology in sustainable supplier evaluation and selection which asked for an analysis of bibliometric data on DEA articles in the selection of sustainable suppliers in peer-reviewed journals listed in the Scopus and Elsevier Web of Science databases. This research considered a range of 87 articles published over a period of 13 years (from 2010 to 2022) (Appendix A) and, thus, the state of the art on this subject could be properly mapped. This mapping considered bibliometrics, scope, the DEA models and extensions used, and the interfaces with other techniques and methodologies. The findings of this research indicate that the impact of DEA in the selection of sustainable suppliers on the supply chain is on the verge to disrupt current supply chain processes and overall business processes in companies affecting its sustainability on market. Moreover, this is the research area that needs more empirical evidence, theoretical and quantitative evidence, and presents a scope for many studies to be carried out in the future. DEA in selection of sustainable suppliers brings many challenges to managing the supply chain where supply chain operations will change accordingly with the use of DEA. The main question for researchers and managers remains how to integrate and manage DEA methodology in the selection of sustainable suppliers as an efficient and effective way in order to optimize the whole supply chain of the organization.

The findings reveal the top contributing authors (Saen R.F., Izadikhah M., Yousefi S., and Azadi M.), the top contributing journals (International Journal of Production Economics, Journal of Cleaner Production and Sustainability-Switzerland), the top contributing countries (Iran, China, United States, India, Taiwan, Hungary, Malaysia, Australia, Germany, and Canada), the top contributing affiliations (Islamic Azad University, North Carolina State University, Indian Institute of Technology, Corvinus University of Budapest, and the National Institute of Industrial Engineering), and present network visualizations of the countries and authors using the VOSviewer software. Moreover, a word cloud with the most frequently used keywords was presented. It should be noted that literature in DEA applied to the process of selection of sustainable suppliers relies on a small number of authors and journals. Namely, studies revolving around sustainable supplier selection with the application of DEA are particularly concentrated and result from the work of a limited academic research network (as shown in Figures 5 and 6). On top of that, one author (Saen R.F.) represents more than 20% of all the publications. As for the analyzed geographic areas, around 70% of all the published papers focus on Iran.

The bibliometric review shows that the use of DEA in sustainable supplier evaluation and selection has shown an increasing trend in the past decade. However, the number of surveyed 87 papers is relatively modest, and even though most of them developed models with practical implications, the results are in line with [49], who suggest that "only a small part of this articles presented a practical application, suggesting that the adoption of DEA is still low in the industry".

This research, however, is not without limitations. Although three researchers were included in the content analysis, the relevance criterion used for the survey analysis of the papers remained somewhat subjective. Moreover, even though the two most reputable scientific databases (Web of Science and Scopus) were consulted, there may have been relevant studies and papers that were not indexed in the abovementioned databases that would have otherwise enriched the study.

In conclusion, there are many opportunities for future investigation and methodological developments of the DEA in many areas related to this research. By analyzing the relevant scientific literature and by the results of bibliometric research, it is possible to conclude that there is currently insufficient awareness and use of DEA in the selection of sustainable suppliers, but through the evaluation of 87 papers, the recognition and appreciation of DEA are visible, which should ultimately lead to its increasing application. Finally, this review depicts directions which should be investigated in future work (the DEA methodology could be used in combination with other models such as GDEA, AHP, TOPSIS, and ANN MCDM models), and the present review provides a sound and solid database for such future studies.

**Author Contributions:** Conceptualization, K.F.Č. and I.M.; methodology, K.F.Č. and I.M.; software, K.F.Č.; validation, K.F.Č. and I.M.; formal analysis, K.F.Č. and I.M.; investigation, K.F.Č. and I.M.; resources, I.M.; data curation, K.F.Č.; writing—original draft preparation, K.F.Č. and I.M.; writing—review and editing, K.F.Č. and I.M.; visualization, J.L.; supervision, K.F.Č., I.M. and J.L.; project administration, I.M. and J.L. All authors have read and agreed to the published version of the manuscript.

**Funding:** The APC was funded by University North, Republic of Croatia.

**Institutional Review Board Statement:** Not applicable.

**Informed Consent Statement:** Not applicable.

**Data Availability Statement:** Not applicable.

**Conflicts of Interest:** The authors declare no conflict of interest.

## Appendix A. List of Selected Papers for Bibliometric Literature Review

| | |
|---|---|
| 1. | Aggarwal, I. Gunreddy, N., Rajan, A.J. A Hybrid Supplier Selection Approach Using Machine Learning and Data Envelopment Analysis, 2021 Innovations in Power and Advanced Computing Technologies (i-PACT), 2021, pp. 1–5, doi: 10.1109/i-PACT52855.2021.9696826. |
| 2. | Amindoust, A. (2018). A resilient-sustainable based supplier selection model using a hybrid intelligent method. Computers and Industrial Engineering, 126, 122–135. doi:10.1016/j.cie.2018.09.031 |
| 3. | Amindoust, A. (2018). Supplier selection considering sustainability measures: An application of weight restriction fuzzy-DEA approach. RAIRO—Operations Research, 52(3), 981–1001. doi:10.1051/ro/2017033 |
| 4. | Amindoust, A., Ahmed, S., & Saghafinia, A. (2012). Supplier performance measurement of palm oil industries from a sustainable point of view in malaysia. BioTechnology: An Indian Journal, 6(6), 155–158. Retrieved from www.scopus.com |
| 5. | Amindoust, A., Ahmed, S., & Saghafinia, A. (2013). Using data envelopment analysis for green supplier selection in manufacturing under vague environment doi:10.4028/www.scientific.net/AMR.622-623.1682 Retrieved from www.scopus.com |
| 6. | Azadi, M., Izadikhah, M., Ramezani, F., & Khadeer, F. (2020). A mixed ideal and anti-ideal DEA model: An application to evaluate cloud service providers. IMA Journal of Management Mathematics, 31(2), 233–256. doi:10.1093/imaman/dpz012 |
| 7. | Azadi, M., Jafarian, M., Saen, R. F., & Mirhedayatian, S. M. (2015). A new fuzzy DEA model for evaluation of efficiency and effectiveness of suppliers in sustainable supply chain management context. Computers and Operations Research, 54, 274–285. doi:10.1016/j.cor.2014.03.002 |
| 8. | Azadi, M., Mirhedayatian, S. M., Saen, R. F., Hatamzad, M., & Momeni, E. (2017). Green supplier selection: A novel fuzzy double frontier data envelopment analysis model to deal with undesirable outputs and dual-role factors. International Journal of Industrial and Systems Engineering, 25(2), 160–181. doi:10.1504/IJISE.2017.081516 |
| 9. | Bai, C., & Sarkis, J. (2014). Determining and applying sustainable supplier key performance indicators. Supply Chain Management, 19(3), 275–291. doi:10.1108/SCM-12-2013-0441 |
| 10. | Bajec, P., Tuljak-Suban, D., & Zalokar, E. (2021). A distance-based AHP-DEA super-efficiency approach for selecting an electric bike sharing system provider: One step closer to sustainability and a win–win effect for all target groups. Sustainability (Switzerland), 13(2), 1–24. doi:10.3390/su13020549 |
| 11. | Boudaghi, E., & Farzipoor Saen, R. (2018). Developing a novel model of data envelopment analysis–discriminant analysis for predicting group membership of suppliers in sustainable supply chain. Computers and Operations Research, 89, 348–359. doi:10.1016/j.cor.2017.01.006 |
| 12. | Chang, K. (2021). A novel contractor selection technique using the extended PROMETHEE II method. Mathematical Problems in Engineering, 2021 doi:10.1155/2021/3664709 |
| 13. | Cheaitou, A., Larbi, R., & Al Housani, B. (2019). Decision making framework for tender evaluation and contractor selection in public organizations with risk considerations. Socio-Economic Planning Sciences, 68 doi:10.1016/j.seps.2018.02.007 |
| 14. | Choudhury, N., Raut, R. D., Gardas, B. B., Kharat, M. G., & Ichake, S. (2018). Evaluation and selection of third party logistics services providers using data envelopment analysis: A sustainable approach. International Journal of Business Excellence, 14(4), 427–453. doi:10.1504/IJBEX.2018.090311 |
| 15. | Dania, W. A. P., Sitepu, I. B. B., & Rucitra, A. L. (2021). Collaboration quality assessment in the sustainable rice supply chain by using an integrated model of QFD-FANP-DEA: A case study of the rice industry in malang. Paper presented at the IOP Conference Series: Earth and Environmental Science, 733(1) doi:10.1088/1755-1315/733/1/012041 Retrieved from www.scopus.com |
| 16. | Dania, W. A. P., Xing, K., & Amer, Y. (2022). The assessment of collaboration quality: A case of sugar supply chain in indonesia. International Journal of Productivity and Performance Management, 71(2), 504–539. doi:10.1108/IJPPM-11-2019-0527 |
| 17. | Dobos, I., & Vörösmarty, G. (2014). Green supplier selection and evaluation using DEA-type composite indicators. International Journal of Production Economics, 157(1), 273–278. doi:10.1016/j.ijpe.2014.09.026 |
| 18. | Dobos, I., & Vörösmarty, G. (2021). Green supplier selection using a common weights analysis of DEA and EOQ types of order allocation. Managerial and Decision Economics, 42(3), 612–621. doi:10.1002/mde.3258 |
| 19. | Ershadi, M. J., Qhanadi Taghizadeh, O., & Hadji Molana, S. M. (2021). Selection and performance estimation of green lean six sigma projects: A hybrid approach of technology readiness level, data envelopment analysis, and ANFIS. Environmental Science and Pollution Research, 28(23), 29394–29411. doi:10.1007/s11356-021-12595-5 |

| 20. | Fakhrzad, M. B., & Nasrollahi, S. (2018). A developed model of data envelopment analysis-discriminant analysis for predicting group membership of suppliers in green supply chain. International Journal of Value Chain Management, 9(4), 378–392. doi:10.1504/IJVCM.2018.095278 |
|---|---|
| 21. | Fallahpour, A., Olugu, E. U., Musa, S. N., Khezrimotlagh, D., & Wong, K. Y. (2016). An integrated model for green supplier selection under fuzzy environment: Application of data envelopment analysis and genetic programming approach. Neural Computing and Applications, 27(3), 707–725. doi:10.1007/s00521-015-1890-3 |
| 22. | Hatami-Marbini, A., Agrell, P. J., Tavana, M., & Khoshnevis, P. (2017). A flexible cross-efficiency fuzzy data envelopment analysis model for sustainable sourcing. Journal of Cleaner Production, 142, 2761–2779. doi:10.1016/j.jclepro.2016.10.192 |
| 23. | Hekmat, S., Amiri, M., & Madraki, G. (2021). Strategic supplier selection in payment industry: A multi-criteria solution for insufficient and interrelated data sources. International Journal of Information Technology and Decision Making, 20(6), 1711–1745. doi:10.1142/S0219622021500474 |
| 24. | Huang, Y., Wang, Y., & Lin, J. (2019). Two-stage fuzzy cross-efficiency aggregation model using a fuzzy information retrieval method. International Journal of Fuzzy Systems, 21(8), 2650–2666. doi:10.1007/s40815-019-00733-8 |
| 25. | Izadikhah, M., & Farzipoor Saen, R. (2020). Ranking sustainable suppliers by context-dependent data envelopment analysis. Annals of Operations Research, 293(2), 607–637. doi:10.1007/s10479-019-03370-4 |
| 26. | Izadikhah, M., & Farzipoor Saen, R. (2019). Solving voting system by data envelopment analysis for assessing sustainability of suppliers. Group Decision and Negotiation, 28(3), 641–669. doi:10.1007/s10726-019-09616-7 |
| 27. | Izadikhah, M., Farzipoor Saen, R., & Ahmadi, K. (2017). How to assess sustainability of suppliers in volume discount context? A new data envelopment analysis approach. Transportation Research Part D: Transport and Environment, 51, 102–121. doi:10.1016/j.trd.2016.11.030 |
| 28. | Izadikhah, M., Farzipoor Saen, R., Ahmadi, K., & Shamsi, M. (2020). How to use fuzzy screening system and data envelopment analysis for clustering sustainable suppliers? A case study in iran. Journal of Enterprise Information Management, 34(1), 199–229. doi:10.1108/JEIM-09-2019-0262 |
| 29. | Izadikhah, M., Saen, R. F., & Ahmadi, K. (2017). How to assess sustainability of suppliers in the presence of dual-role factor and volume discounts? A data envelopment analysis approach. Asia-Pacific Journal of Operational Research, 34(3) doi:10.1142/S0217595917400164 |
| 30. | Izadikhah, M., Saen, R. F., & Roostaee, R. (2018). How to assess sustainability of suppliers in the presence of volume discount and negative data in data envelopment analysis? Annals of Operations Research, 269(1–2), 241–267. doi:10.1007/s10479-018-2790-6 |
| 31. | Jafarzadeh Ghoushchi, S., Dodkanloi Milan, M., & Jahangoshai Rezaee, M. (2018). Evaluation and selection of sustainable suppliers in supply chain using new GP-DEA model with imprecise data. Journal of Industrial Engineering International, 14(3), 613–625. doi:10.1007/s40092-017-0246-2 |
| 32. | Jain, V., Kumar, S., Kumar, A., & Chandra, C. (2016). An integrated buyer initiated decision-making process for green supplier selection. Journal of Manufacturing Systems, 41, 256–265. doi:10.1016/j.jmsy.2016.09.004 |
| 33. | Jauhar, S. K., Amin, S. H., & Zolfagharinia, H. (2021). A proposed method for third-party reverse logistics partner selection and order allocation in the cellphone industry. Computers and Industrial Engineering, 162 doi:10.1016/j.cie.2021.107719 |
| 34. | Jauhar, S. K., & Pant, M. (2016). Using differential evolution to develop a carbon-integrated model for performance evaluation and selection of sustainable suppliers in indian automobile supply chain doi:10.1007/978-981-10-0451-3_47 Retrieved from www.scopus.com |
| 35. | Jauhar, S. K., Pant, M., & Nagar, M. C. (2015). Differential evolution for sustainable supplier selection in pulp and paper industry: A DEA based approach. Computer Methods in Materials Science, 15(1), 118–126. Retrieved from www.scopus.com |
| 36. | Karimi, A., Jafarzadeh-Ghoushchi, S., & Mohtadi-Bonab, M. A. (2020). Presenting a new model for performance measurement of the sustainable supply chain of shoa panjereh company in different provinces of iran (case study). International Journal of Systems Assurance Engineering and Management, 11(1), 140–154. doi:10.1007/s13198-019-00932-4 |
| 37. | Karimi, B., Azadi, M., Farzipoor Saen, R., & Fosso Wamba, S. (2022). Theory of binary-valued data envelopment analysis: An application in assessing the sustainability of suppliers. Industrial Management and Data Systems, 122(3), 682–701. doi:10.1108/IMDS-09-2021-0555 |
| 38. | Kaur, H., & Prakash Singh, S. (2021). Multi-stage hybrid model for supplier selection and order allocation considering disruption risks and disruptive technologies. International Journal of Production Economics, 231 doi:10.1016/j.ijpe.2020.107830 |

| 39. | Kumar, A., Jain, V., & Kumar, S. (2014). A comprehensive environment friendly approach for supplier selection. Omega (United Kingdom), 42(1), 109–123. doi:10.1016/j.omega.2013.04.003 |
|---|---|
| 40. | Kumar, A., Jain, V., Kumar, S., & Chandra, C. (2016). Green supplier selection: A new genetic/immune strategy with industrial application. Enterprise Information Systems, 10(8), 911–943. doi:10.1080/17517575.2014.986220 |
| 41. | Kuo, R. J., Wang, Y. C., & Tien, F. C. (2010). Integration of artificial neural network and MADA methods for green supplier selection. Journal of Cleaner Production, 18(12), 1161–1170. doi:10.1016/j.jclepro.2010.03.020 |
| 42. | Li, F., Deng, L., Li, L., Cheng, Z., & Yu, H. (2020). A two-stage model for monitoring the green supplier performance considering dual-role and undesirable factors. Asia Pacific Journal of Marketing and Logistics, 32(1), 253–280. doi:10.1108/APJML-02-2019-0110 |
| 43. | Li, F., Wu, L., Zhu, Q., Yu, Y., Kou, G., & Liao, Y. (2020). An eco-inefficiency dominance probability approach for chinese banking operations based on data envelopment analysis. Complexity, 2020 doi:10.1155/2020/3780232 |
| 44. | Mahdiloo, M., Saen, R. F., & Lee, K. (2015). Technical, environmental and eco-efficiency measurement for supplier selection: An extension and application of data envelopment analysis. International Journal of Production Economics, 168, 279–289. doi:10.1016/j.ijpe.2015.07.010 |
| 45. | Mahmoudi, A., Abbasi, M., & Deng, X. (2022). Evaluating the performance of the suppliers using hybrid DEA-OPA model: A sustainable development perspective. Group Decision and Negotiation, 31(2), 335–362. doi:10.1007/s10726-021-09770-x |
| 46. | Moghaddas, Z., Tosarkani, B. M., & Yousefi, S. (2022). A developed data envelopment analysis model for efficient sustainable supply chain network design. Sustainability (Switzerland), 14(1) doi:10.3390/su14010262 |
| 47. | Moheb-Alizadeh, H., & Handfield, R. (2018). An integrated chance-constrained stochastic model for efficient and sustainable supplier selection and order allocation. International Journal of Production Research, 56(21), 6890–6916. doi:10.1080/00207543.2017.1413258 |
| 48. | Moheb-Alizadeh, H., & Handfield, R. (2019). Sustainable supplier selection and order allocation: A novel multi-objective programming model with a hybrid solution approach. Computers and Industrial Engineering, 129, 192–209. doi:10.1016/j.cie.2019.01.011 |
| 49. | Moheb-Alizadeh, H., Handfield, R., & Warsing, D. (2021). Efficient and sustainable closed-loop supply chain network design: A two-stage stochastic formulation with a hybrid solution methodology. Journal of Cleaner Production, 308 doi:10.1016/j.jclepro.2021.127323 |
| 50. | Momeni, M., & Vandchali, H. R. (2017). Providing a structured methodology for supplier selection and evaluation for strategic outsourcing. International Journal of Business Performance and Supply Chain Modelling, 9(1), 66–85. doi:10.1504/IJBPSCM.2017.083888 |
| 51. | Nemati, M., Farzipoor Saen, R., & Matin, R. K. (2021). A data envelopment analysis approach by partial impacts between inputs and desirable-undesirable outputs for sustainable supplier selection problem. Industrial Management and Data Systems, 121(4), 809–838. doi:10.1108/IMDS-12-2019-0653 |
| 52. | Ngobeni, V., & Breitenbach, M. C. (2021). Production and scale efficiency of south african water utilities: The case of water boards. Water Policy, 23(4), 862–879. doi:10.2166/wp.2021.055 |
| 53. | Nikabadi, M. S., & Moghaddam, H. F. (2021). An integrated approach of adaptive neuro-fuzzy inference system and dynamic data envelopment analysis for supplier selection. International Journal of Mathematics in Operational Research, 18(4), 503–527. doi:10.1504/IJMOR.2021.114206 |
| 54. | Nikfarjam, H., Rostamy-Malkhalifeh, M., & Noura, A. (2018). A new robust dynamic data envelopment analysis approach for sustainable supplier evaluation. Advances in Operations Research, 2018 doi:10.1155/2018/7625025 |
| 55. | Pishchulov, G., Trautrims, A., Chesney, T., Gold, S., & Schwab, L. (2019). The voting analytic hierarchy process revisited: A revised method with application to sustainable supplier selection. International Journal of Production Economics, 211, 166–179. doi:10.1016/j.ijpe.2019.01.025 |
| 56. | Rajak, S., Parthiban, P., & Dhanalakshmi, R. (2021). A DEA model for evaluation of efficiency and effectiveness of sustainable transportation systems: A supply chain perspective. International Journal of Logistics Systems and Management, 40(2), 220–241. doi:10.1504/IJLSM.2021.118737 |
| 57. | Rashidi, K. (2020). AHP versus DEA: A comparative analysis for the gradual improvement of unsustainable suppliers. Benchmarking, 27(8), 2283–2321. doi:10.1108/BIJ-11-2019-0505 |
| 58. | Rashidi, K., & Cullinane, K. (2019). A comparison of fuzzy DEA and fuzzy TOPSIS in sustainable supplier selection: Implications for sourcing strategy. Expert Systems with Applications, 121, 266–281. doi:10.1016/j.eswa.2018.12.025 |

| 59. | Raut, R., Kharat, M., Kamble, S., & Kumar, C. S. (2018). Sustainable evaluation and selection of potential third-party logistics (3PL) providers: An integrated MCDM approach. Benchmarking, 25(1), 76–97. doi:10.1108/BIJ-05-2016-0065 |
|---|---|
| 60. | Ryu, Y., & Sueyoshi, T. (2021). Examining the relationship between the economic performance of technology-based small suppliers and socially sustainable procurement. Sustainability (Switzerland), 13(13) doi:10.3390/su13137220 |
| 61. | Shabanpour, H., Fathi, A., Yousefi, S., & Saen, R. F. (2019). Ranking sustainable suppliers using congestion approach of data envelopment analysis. Journal of Cleaner Production, 240 doi:10.1016/j.jclepro.2019.118190 |
| 62. | Shabanpour, H., Yousefi, S., & Farzipoor Saen, R. (2017). Future planning for benchmarking and ranking sustainable suppliers using goal programming and robust double frontiers DEA. Transportation Research Part D: Transport and Environment, 50, 129–143. doi:10.1016/j.trd.2016.10.022 |
| 63. | Shabanpour, H., Yousefi, S., & Saen, R. F. (2017). Forecasting efficiency of green suppliers by dynamic data envelopment analysis and artificial neural networks. Journal of Cleaner Production, 142, 1098–1107. doi:10.1016/j.jclepro.2016.08.147 |
| 64. | Sharafi, H., Soltanifar, M., & Lotfi, F. H. (2022). Selecting a green supplier utilizing the new fuzzy voting model and the fuzzy combinative distance-based assessment method. EURO Journal on Decision Processes, 10 doi:10.1016/j.ejdp.2021.100010 |
| 65. | Shi, P., Yan, B., Shi, S., & Ke, C. (2015). A decision support system to select suppliers for a sustainable supply chain based on a systematic DEA approach. Information Technology and Management, 16(1), 39–49. doi:10.1007/s10799-014-0193-1 |
| 66. | Soltanifar, M., & Sharafi, H. (2022). A modified DEA cross efficiency method with negative data and its application in supplier selection. Journal of Combinatorial Optimization, 43(1), 265–296. doi:10.1007/s10878-021-00765-7 |
| 67. | Tavana, M., Nazari-Shirkouhi, S., & Farzaneh Kholghabad, H. (2021). An integrated quality and resilience engineering framework in healthcare with Z-number data envelopment analysis. Health Care Management Science, 24(4), 768–785. doi:10.1007/s10729-021-09550-8 |
| 68. | Tavana, M., Shabanpour, H., Yousefi, S., & Farzipoor Saen, R. (2017). A hybrid goal programming and dynamic data envelopment analysis framework for sustainable supplier evaluation. Neural Computing and Applications, 28(12), 3683–3696. doi:10.1007/s00521-016-2274-z |
| 69. | Tavassoli, M., Saen, R. F., & Zanjirani, D. M. (2020). Assessing sustainability of suppliers: A novel stochastic-fuzzy DEA model. Sustainable Production and Consumption, 21, 78–91. doi:10.1016/j.spc.2019.11.001 |
| 70. | Torres-Ruiz, A., & Ravindran, A. R. (2019). Use of interval data envelopment analysis, goal programming and dynamic eco-efficiency assessment for sustainable supplier management. Computers and Industrial Engineering, 131, 211–226. doi:10.1016/j.cie.2019.02.008 |
| 71. | Tsai, C., Lee, H., & Gan, G. (2021). A new fuzzy dea model for solving the mcdm problems in supplier selection. Journal of Marine Science and Technology (Taiwan), 29(1), 89–95. doi:10.51400/2709-6998.1006 |
| 72. | Vörösmarty, G., & Dobos, I. (2020). A literature review of sustainable supplier evaluation with data envelopment analysis. Journal of Cleaner Production, 264 doi:10.1016/j.jclepro.2020.121672 |
| 73. | Vörösmarty, G., & Dobos, I. (2019). Supplier evaluation with environmental aspects and common DEA weights. Periodica Polytechnica Social and Management Sciences, 27(1), 17–25. doi:10.3311/PPso.11814 |
| 74. | Wang, C., Ho, H. T., Luo, S., & Lin, T. (2017). An integrated approach to evaluating and selecting green logistics providers for sustainable development. Sustainability (Switzerland), 9(2) doi:10.3390/su9020218 |
| 75. | Wang, C., Nguyen, V. T., Thai, H. T. N., Tran, N. N., & Tran, T. L. A. (2018). Sustainable supplier selection process in edible oil production by a hybrid fuzzy analytical hierarchy process and green data envelopment analysis for the SMEs food processing industry. Mathematics, 6(12) doi:10.3390/math6120302 |
| 76. | Wang, C, Pham, T. T., & Nhieu, N. (2021). Multi-layer fuzzy sustainable decision approach for outsourcing manufacturer selection in apparel and textile supply chain. Axioms, 10(4) doi:10.3390/axioms10040262 |
| 77. | Wang, C, Tsai, H., Ho, T., Nguyen, V., & Huang, Y. (2020). Multi-criteria decision making (MCDM) model for supplier evaluation and selection for oil production projects in vietnam. Processes, 8(2) doi:10.3390/pr8020134 |
| 78. | Wu, M, Zhang, C., Liu, X., & Fan, J. (2019). Green supplier selection based on DEA model in interval-valued pythagorean fuzzy environment. IEEE Access, 7, 108001–108013. doi:10.1109/ACCESS.2019.2932770 |
| 79. | Yan, X., Bao, X., Zhao, R., & Li, F. (2022). Performance measurement for green supplier selection based on data envelopment analysis. Environmental Science and Pollution Research, doi:10.1007/s11356-021-17897-2 |
| 80. | Yousefi, S., & Mohamadpour Tosarkani, B. (2022). An analytical approach for evaluating the impact of blockchain technology on sustainable supply chain performance. International Journal of Production Economics, 246 doi:10.1016/j.ijpe.2022.108429 |

| 81. | Yu, M. -Chun, & Su, M. -Hong. (2017). Using fuzzy DEA for green suppliers selection considering carbon footprints. Sustainability (Switzerland), 9(4) doi:10.3390/su9040495 |
|---|---|
| 82. | Zarbakhshnia, N., & Jaghdani, T. J. (2018). Sustainable supplier evaluation and selection with a novel two-stage DEA model in the presence of uncontrollable inputs and undesirable outputs: A plastic case study. International Journal of Advanced Manufacturing Technology, 97(5–8), 2933–2945. doi:10.1007/s00170-018-2138-z |
| 83. | Zhang, Z., & Liao, H. (2022). A stochastic cross-efficiency DEA approach based on the prospect theory and its application in winner determination in public procurement tenders. Annals of Operations Research, doi:10.1007/s10479-022-04539-0 |
| 84. | Zhao, S., Wang, J., Ye, M., Huang, Q., & Si, X. (2022). An evaluation of supply chain performance of China's prefabricated building from the perspective of sustainability. Sustainability (Switzerland), 14(3) doi:10.3390/su14031299 |
| 85. | Zhou, X., Li, L., Wen, H., Tian, X., Wang, S., & Lev, B. (2021). Supplier's goal setting considering sustainability: An uncertain dynamic data envelopment analysis based benchmarking model. Information Sciences, 545, 44–64. doi:10.1016/j.ins.2020.07.074 |
| 86. | Zhou, X., Pedrycz, W., Kuang, Y., & Zhang, Z. (2016). Type-2 fuzzy multi-objective DEA model: An application to sustainable supplier evaluation. Applied Soft Computing Journal, 46, 424–440. doi:10.1016/j.asoc.2016.04.038 |
| 87. | Zoroofchi, K. H., Saen, R. F., Lari, P. B., & Azadi, M. (2018). A combination of range-adjusted measure, cross-efficiency and assurance region for assessing sustainability of suppliers in the presence of undesirable factors. International Journal of Industrial and Systems Engineering, 29(2), 163–176. doi:10.1504/IJISE.2018.091898 |

Refs [15,49,58–121] are part of Appendix A.

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
