# Peer review of "Application of Data Envelopment Analysis (DEA) in the Selection of Sustainable Suppliers: A Review and Bibliometric Analysis"

_sustainability, doi:10.3390/su14116672_

Round 1
Reviewer 1 Report
The manuscript is well-written, and the topic is interesting. However, the following concern should be addressed:
- The DEA is combined with the MCDM methods several times to improve the performance of the DEA model. The authors should clearly provide a table and discuss the studies in this field of study.
- Why the authors use VOSviewer software? Why did they not select Citespace software which seems to be better?
- Why did the authors repeat the Data Envelopment Analysis (DEA) several times? Just for the first time is enough. Also, check for other terms.
- In the introduction section, the authors mentioned several contributions. In my opinion, some of which are not fair. The authors should mention the fact, some of which cannot be a novelty.
- Please recheck Figures 6 and 7. Are you sure about the data?
Author Response
Dear Reviewer,
We would like to thank you for all your comments regarding our paper that contributed to its improvement.
Our answers are given in the PDF below.
Kind regards,
The Authors

Reviewer 2 Report
This paper is a review paper on supplier selection for which a searching algorithm is proposed according to Scopus information. I would like to recommend adding some influential articles about the main idea in the Introduction. Please consider the following two articles.
1) A Developed Data Envelopment Analysis Model for Efficient Sustainable
Supply Chain Network Design Z Moghaddas, BM Tosarkani, S Yousefi
Sustainability 14 (1), 262.
2) A multi-criteria intuitionistic fuzzy group decision-making method
for supplier selection with the VIKOR method. Razieh Roostaee, Mohammad
Izadikhah, Farhad Hosseinzadeh Lotfi, Mohsen Rostamy-Malkhalifeh.
International journal of fuzzy system applications (IJFSA).
Author Response

(The authors gave the same response as above.)

Reviewer 3 Report
1. Selection of Sustainable Suppliers: Bibliometric literature review of Data Envelopment Analysis (DEA) applications- Authors are not conducting the evaluation of any suppliers, so how they can write in the title of the paper selection of sustainable suppliers.
2. The authors are showing 87 papers they have reviewed in figure 1 and Appendix A. but in the reference section, they are showing only 44 papers. how did this mismatch happen?
Author Response

(The authors gave the same response as above.)

Reviewer 4 Report
The subject is within the scope of the journal and the objective of the research is well stated.
The presented results are interesting. However, the work needs to:
- better highlight the motivation and innovation of this work
- enlarge the literature review in this survey
- improve the critical discussion of the selected documents
- further elaborate on the evidence of the research and future research paths.
Consider the following documents:
Design of modern supply chain networks using fuzzy bargaining game and data envelopment analysis. IEEE Transactions on Automation Science and Engineering
A game-theoretical design technique for multi-stage supply chains under uncertainty. In 2018 IEEE 14th International Conference on Automation Science and Engineering
Intermodal terminal planning by petri nets and data envelopment analysis. Control Engineering Practice
A decision making procedure for robust train rescheduling based on mixed integer linear programming and Data Envelopment Analysis. Applied Mathematical Modelling
Author Response

(The authors gave the same response as above.)

Round 2
Reviewer 1 Report
The manuscript can be accepted.
Author Response
Thank you so much for your valuable insights, comments and suggestions!
The Authors
Reviewer 3 Report
I can see some improvement in the manuscript after revision. but still, the authors have failed to answer my previous comments.
Authors should change the title of the paper as per their work. in my point of view, it is not reflected at present.
Authors should cite all 87 papers in the reference section.
The authors should redesign their abstract to answer the following questions: What did you do? Why did you do it? What did you learn? What do you now recommend based on your work?
The introduction section is poorly organized; there are lots of missing links, and in addition, the problem should be explained based on the necessity of researching the current subject in the introductory section, which has not happened in the current format. This section should be completely revised.
The authors should mention the research question and research objective in the Introduction section.
The literature review is also poorly organised. I have never seen any research gap as a sub-section, which is very important for proving the novelty of the paper. If possible try to add a research question in the Introduction section, which may assist the readers to catch the point of the paper in the first section itself.
1. Conclusions are very vague and general; they should list and comments on some of the specific basic findings/results such as describe and discuss “weaker areas”, detail needs to be furnished.
Totally, the paper needs complete elaboration. It is not at the level of academic journals in both structure and content. The contribution is minor and the presentations of the work are not convincing.
Author Response
Dear Reviewer,
We have addressed all the issues raised by you in the first and second round of revision. We hereby want to express our immense gratitude for your most valuable comments and suggestions. We strongly believe your comments have considerably improved our manuscript.
Please do not hesitate to contact us if you believe we can further improve our paper.
Kindest regards from Croatia,
The Authors
